# Plasticity of both planar cell polarity and cell identity during the development of *Drosophila*

**Pedro Saavedra[1]\*, Jean-Paul Vincent[2], Isabel M Palacios[1], Peter A Lawrence[1], José Casal[1]\***

[1]Department of Zoology, University of Cambridge, Cambridge, United Kingdom; [2]MRC National Institute for Medical Research, London, United Kingdom

**Abstract** *Drosophila* has helped us understand the genetic mechanisms of pattern formation. Particularly useful have been those organs in which different cell identities and polarities are displayed cell by cell in the cuticle and epidermis (*Lawrence, 1992*; *Bejsovec and Wieschaus, 1993*; *Freeman, 1997*). Here we use the pattern of larval denticles and muscle attachments and ask how this pattern is maintained and renewed over the larval moult cycles. During larval growth each epidermal cell increases manyfold in size but neither divides nor dies. We follow individuals from moult to moult, tracking marked cells and find that, as cells are repositioned and alter their neighbours, their identities change to compensate and the pattern is conserved. Single cells adopting a new fate may even acquire a new polarity: an identified cell that makes a forward-pointing denticle in the first larval stage may make a backward-pointing denticle in the second and third larval stages.

## Introduction

There are three larval stages in the development of *Drosophila*, L1–L3 (*Szabad et al., 1979*; *Dambly-Chaudière and Ghysen, 1986*; *Campos-Ortega and Hartenstein, 1997*). Ventrally, each segment of the abdominal epidermis has a belt of thorny denticles (*Lohs-Schardin et al., 1979*; *Dougan and DiNardo, 1992*; *Campos-Ortega and Hartenstein, 1997*; *Payre et al., 1999*). Each belt is built by seven lines of cells (*Figure 1*) that together produce seven imperfectly defined rows of denticles (*Price et al., 2006*; *Walters et al., 2006*); denticle rows 0 and 1 are made by cells of the posterior compartment (P) (*Dougan and DiNardo, 1992*) and rows 2–7 by the anterior compartment (A). The denticles in rows 2, 3, 5, 6 and 7 point backwards while those in rows 0, 1 and 4 point forwards (*Lohs-Schardin et al., 1979*). The cells that make denticle rows 2 and 5 in the embryo have also an additional function: they are the tendon cells that link epidermal cells to muscles (*Figure 1C–E*; *Hatini and DiNardo, 2001*; *Volohonsky et al., 2007*; reviewed in *Volk (1999)*).

During each moult cycle, the larval epidermis secretes a new cuticle under the old one and when this process is completed, the old cuticle is sloughed off. Over the two larval moult cycles the epidermal cells do not change in number, but they undergo endoreplication of their DNA and grow considerably (*Edgar and Orr-Weaver, 2001*). Here we describe how the epidermal cells behave during the three larval stages and ask how the patterns of muscle attachments and cuticular denticles are maintained.

## Results and discussion

### Denticles are formed differently in the embryo and the larva

During embryogenesis, the actin-based pre-denticles, the precursors of the cuticular denticles, are formed temporarily at the apico-posterior limits of the cells and all point backwards (*Figure 1A*; *Dickinson and Thatcher, 1997*). However, by the L1 stage the completed denticles of rows 1 and 4 now

**\*For correspondence:** padgs2@cam.ac.uk (PS); jec85@cam.ac.uk (JC)

**Competing interests:** The authors declare that no competing interests exist.

**Reviewing editor**: Helen McNeill, The Samuel Lunenfeld Research Institute, Canada

**eLife digest** Fly larvae grow in an unusual way. Most embryos grow by increasing the number of cells in the embryo, but fly larvae grow by increasing the size of their cells. The epidermal cells in the growing larvae secrete a hard skin or cuticle that is renewed three times as they grow. This cuticle is decorated with teeth called denticles that the larvae use to grip surfaces as they crawl on them. The denticles are arranged in six rows during all three larval stages.

It has long been assumed that if a cell in the first larval stage makes the denticles belonging to a given row, then the same cell will make denticles in the same row in the second and third larval stages. Now Saavedra et al. report that this assumption is mistaken and that the epidermal cells rearrange extensively between the first and second larval stages, and that cells acquire different identities to keep the pattern constant.

Saavedra et al. marked small groups of cells in the embryo and plotted the positions of these cells as the larvae progressed through the three stages of development. These measurements showed that as the larvae grow, the cells changed their positions relative to each other. This meant that, in order to keep essentially the same pattern of denticle rows, the cells had to change their identity.

Some of these changes were quite dramatic. Consider, for example, the embryonic cells that make the denticles in the second of the six rows during the first larval stage. In the embryo, these cells are tendons and attach to the muscles needed for crawling. Saavedra et al. found that these cells remain attached to the same muscles throughout growth, but that they do not make denticles during the second and third larval stages. Instead, the denticles in the second row of later larval stages are made by other cells, and these new second row cells are not attached to any muscles. In another example of these changes, some cells make denticles that point away from the head during the first larval stage, and then make denticles that point towards the head during later stages. Thus, cells can change both their identity (e.g., whether they are attached to muscles or not) and their orientation (also known as the cell polarity) during the development of a larva.

The work of Saavedra et al. illustrates how organisms adapt developmental mechanisms that have been stabilised for millions of years and for this reason limit the kinds of morphological changes that are possible.

point forwards (*Figure 1E*; *Lohs-Schardin et al., 1979*) and it is not clear when or how this change of polarity occurs. However, our observations suggest that rows 1 and 4 have broader cells and start to behave differently from the other rows shortly before stage 16, at the beginning of cuticle formation (*Figure 2A*).

Actin pre-denticles are also found in the epidermis of the later parts of L1 and L2 as they each begin to construct the subsequent stage (we call such larvae pre-L2 and pre-L3). The pre-denticles do not all point backwards: in both pre-L2 and pre-L3 stages the pre-denticles of rows 1 and 4 point forwards (*Figure 1B*). Row 0 denticles, which are present in only the L2 and L3 stages, also come from pre-denticles that point anteriorly. Therefore, unlike the embryonic pre-denticles, all the larval pre-denticles are an accurate harbinger of the orientation of the denticles that they will make.

## The arrangement of the epithelial cells changes during larval development

The larva grows substantially and moults twice with only subtle changes to the pattern (*Dambly-Chaudière and Ghysen, 1986*; *Hartenstein and Campos-Ortega, 1986*). It has been assumed that the cells become polytene as they do in *Calliphora* (*Pearson, 1974*), and that they neither divide nor die. These assumptions led to the reasonable expectations that the arrangement of the cells as well as their identities are conserved throughout the three larval stages (*Szabad et al., 1979*; *Dambly-Chaudière and Ghysen, 1986*; *Hartenstein and Campos-Ortega, 1986*; *Bate and Martínez Arias, 1993*; *Campos-Ortega and Hartenstein, 1997*). However, as we now demonstrate, both these expectations are mistaken.

In the embryo, the lines of epidermal cells that will produce the denticles of L1 are more tightly compacted than those cells that do not produce denticles (*Price et al., 2006*; *Walters et al., 2006*)

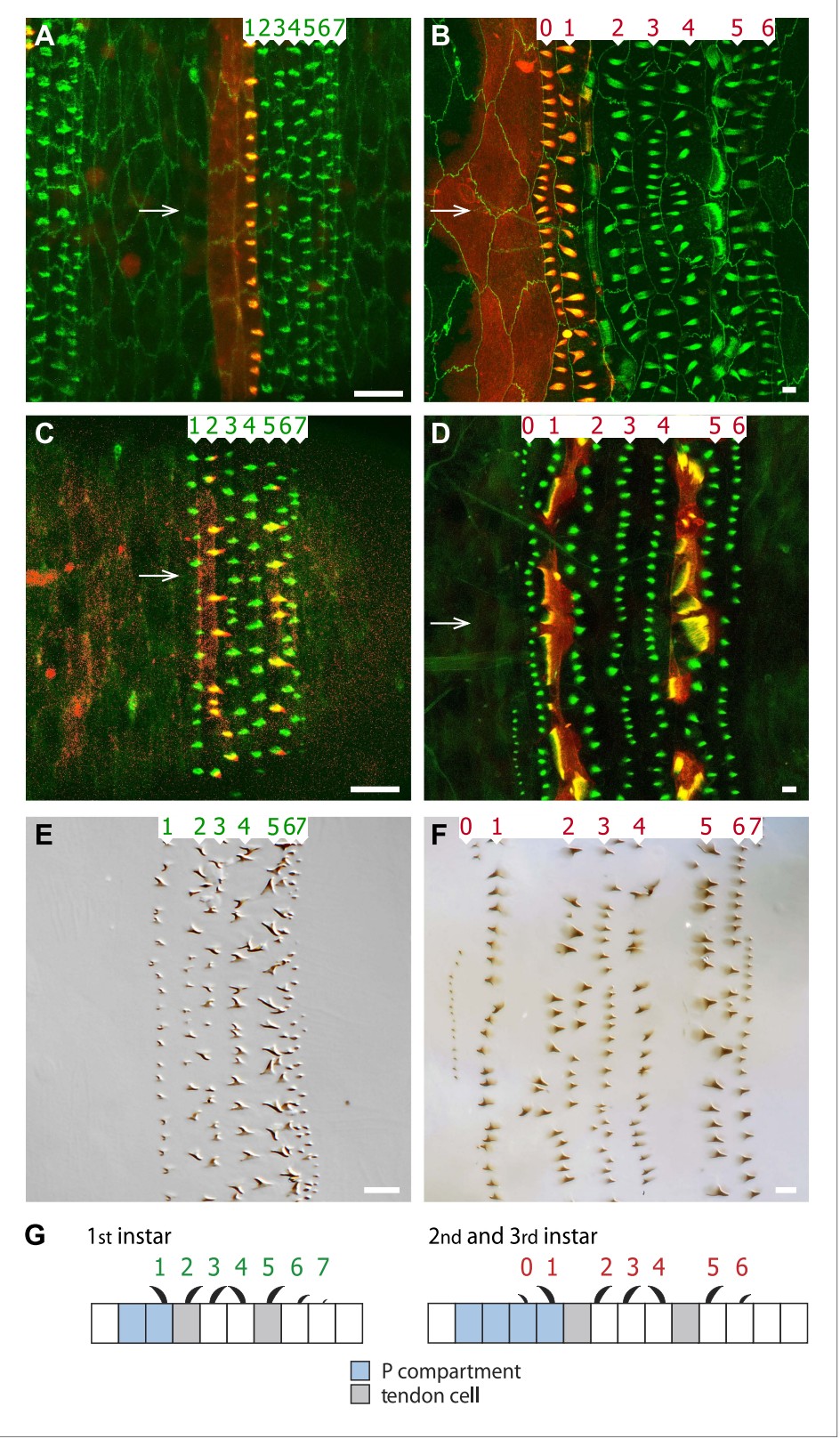

**Figure 1**. The arrangement of epithelial cells differs between embryo and larva. Anterior is to the left in all figures. (**A** and **B**) An embryo is shown in (**A**) and the pre-L3 (i.e., the new third instar epidermis developing under the second instar cuticle) in (**B**). The seven rows of pre-denticles are numbered and arrows indicate the ventral midline. *Figure 1. Continued on next page*

*Figure 1. Continued*

The pre-denticles are labelled with utrp::GFP and the cell outlines with DE-cad::GFP (both in green). Posterior cells and pre-denticles (rows 1 in (**A**) and rows 0 and 1 in (**B**)) show red because *en.GAL4* is driving expression of *UAS. cherry::moesin* in the entire P compartment. (**C** and **D**) Embryo (**C**) and pre-L3 (**D**). Pre-denticles labelled with utrp::GFP as above. The muscle-attaching tendon cells are marked by *sr.GAL4* driving expression of *UAS.cherry::moesin* (red). In the embryo (**C**) *sr.GAL4* marks the pre-denticles of rows 2 and 5, made by the two lines of tendon cells in the embryo. In the pre-L3 larva (**D**) note the actin palisades in the tendon cells that are labelled in both green and red. In the larva, no pre-denticles are made by these two lines of cells. (**E** and **F**) show the cuticular denticles of the L1 (**E**) and L3 (**F**) larvae. Scale bars are 10 μm. (**G**) Diagrams of the embryonic and larval ventral epithelium. The green numbers indicate rows of denticles in L1, the red numbers in L2 and L3. Their polarities are indicated. Note the many changes between embryo and larvae (see also *Figure 2*).

and include tendon cells that themselves make the pre-denticles of rows 2 and 5 (*Figures 1C and 2*; *Hatini and DiNardo, 2001*). Note that in what follows there are generalisations as accurate as we can make them but, in truth, each segment differs slightly from the next. Sometimes the lines of cells and the denticle rows are incomplete or partially duplicated, occasional cells are difficult to allocate or sit in an ambiguous position. The tendon cells are separated by the two lines of cells that will make denticle rows 3 and 4 (*Figure 1C,E*). The embryonic P compartment is two-cells wide in the anteroposterior axis, the posterior of these two lines of cells making row 1 denticles (*Figure 1A,G*; *Dougan and DiNardo, 1992*). In the larva, the arrangement of the cells differs from the embryo in three major respects: first, unlike the tendon cells of the embryo, the tendon cells of the larva do not themselves make denticles. One row of tendon cells is located between denticle rows 1 and 2 and the other between denticle rows 4 and 5 (*Figure 1B,D,F*). Second, in the embryo there are two lines of cells between the tendon cells, while in the larva the tendon cells are separated by three lines of cells. Third, in the embryo, the P compartment is two cells wide, but it becomes about four cells wide in the larva (*Figure 1A,B*). These changes occur prior to the L2 stage and clearly involve a reorganisation of the cells that gives a substantial increase in length, along the anteroposterior axis (*Figure 2*). Nevertheless, in spite of this cell rearrangement, the cuticular pattern is very similar in all the three larval stages (*Figure 1E,F*); suggesting that some cells must be reallocated to different fates during the transition from the embryo to the L2 larva.

We have quantified the arrangement and number of cells, studying individuals during embryogenesis and revisiting the same individuals as pre-L3 larvae. Other individuals were studied as pre-L2 larvae. The number of cells in a defined rectangular portion of the segment remained constant in all three stages at a mean of about 73 cells (*Figure 2C–F*), confirming that the epidermal cells do not divide or die. In the embryo, the average number of lines of cells found in the anteroposterior axis of this portion was about 14 but it increased to 18 in the pre-L2 larva and remained unchanged thereafter and up to the pre-L3 stage (*Figure 2C–F*). Also, the proportions of a fixed rectangular region of the segment changed between embryo and the pre-L3 larva. The ratios of the lengths of the anteroposterior to mediolateral axes were compared; there was a large change in the shape of this rectangle from the embryo to the L2 and L3 larval stages (*Figure 2*). We measured the shape changes separately in the denticulate and naked cuticle; the cells in these two regions rearranged in a similar way, each one increasing by roughly 2 cells in the anteroposterior axis (data not shown).

Thus, both ways of describing the rearrangement of the cells argue that, between the L1 and L2 larval stages, the ventral epithelial cells converge into the midline and extend in the anteroposterior axis. This mode of cell rearrangement is found in many systems and is known as convergent extension (reviewed in *Keller, 2002*; *Wallingford et al., 2002*).

## How exactly do the cells rearrange?

To investigate in more detail how the cells rearrange, we induced clones soon after the time of cellularisation in the embryo and found these usually divided one to two times during late embryogenesis, giving rise to small clones of marked cells (*Table 1*). We then followed individual clones during the next two larval stages (*Figure 3*). To track the cells in the clones we needed to number the lines of cells (in roman numerals) to distinguish them from the rows of denticles that they contribute to (see *Figure 3*, 'Materials and methods'). For example, a marked cell might divide and label three cells that belong to line III in the embryo; all such cells would form pre-denticles of row 3. This clone could then be

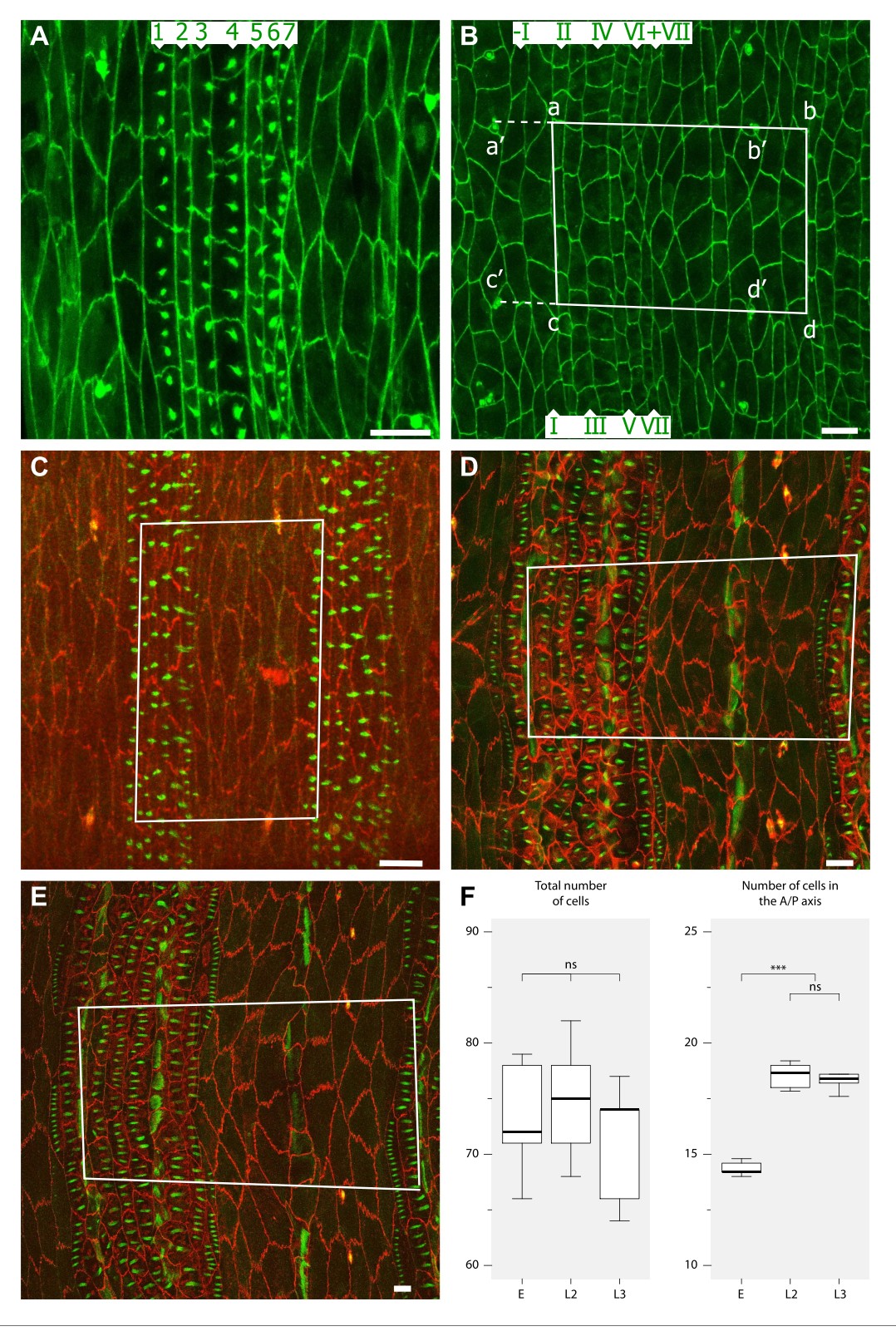

**Figure 2**. Convergent extension in the anteroposterior axis between pre-L1 and L2. (**A**) Mid stage embryo with pre-denticles. The seven rows of pre-denticles (1–7) are indicated. At first all pre-denticles are found on the posterior boundaries of the appropriate rows. However, in later embryos pre-denticle rows 1 and 4, the two rows
*Figure 2. Continued on next page*

*Figure 2. Continued*

that will make denticles pointing forwards, are now situated near the middle of the cells. This suggests that some movement of the pre-denticles may be part of polarity reversal. Also it may be relevant that cell lines I and IV of the embryo are the only lines that make extra lines of cells in the larva, and therefore contribute to convergent extension. Labelling for (**A–E**): The pre-denticles are labelled with utrp::GFP and the cell outlines with DE-cad::GFP (**A** and **B**) or DE-cad::tomato (**C–E**). (**B**) Late stage embryo, before moulting to L1 but after actin pre-denticles have gone. The pattern is similar to the earlier embryo, with two lines of cells between the tendon cells (II and V). (**C**) Mid stage embryo showing the marked portion of the epidermis. The rectangle demarcates a segment in the anteroposterior axis and the region between the pair of ventral sensorial papillae (p1) (***Dambly-Chaudière and Ghysen, 1986***) in the mediolateral axis. The total number of cells and the numbers along the axes were counted, see (**F**). (**D**) The pre-L2 stage. By this stage the cells have rearranged and extended in the anteroposterior axis: the number of cells in that axis has increased from ca 14 to 18 cells per segment. The fixed rectangle has changed shape dramatically but contains the same number of cells as in the embryo (***Table 1***). This change of dimensions is due to convergent extension that involves cell rearrangement as well as alterations in the shapes of cells. (**E**) The pre-L3 stage. The pattern of cells and the shape of the fixed rectangle resembles that in the pre-L2. (**F**) Quantitation of the evidence for convergent extension: Boxplots (***Frigge et al., 1989***) of the number of cells enclosed by fixed pattern landmarks remains the same while these cells extend in the anteroposterior axis and narrow in the mediolateral axis. Pairwise comparisons using t tests with Bonferroni adjustment (ns = not significant, *** = p<0.001).

revisited in the pre-L3 to see if the same three cells were still forming row 3 pre-denticles in the larva and ask whether the disposition of these cells in the larva differed from the earlier arrangement in the embryo.

The number of cells per clone varied between one and four, or rarely five, (***Table 1*** and other clones studied but not included therein) and, for each clone (n = 121), this number did not change significantly between the embryo (n = 404 cells) and the pre-L3 larva (n = 410); confirming that cell divisions and cell death, if any, are rare in the larval epidermis. In our opinion this small inconsistency is due to recording error, caused by weaker staining in the embryo. The lack of any epidermal cell death was confirmed by means of a caspase marker. The Apoliner marker stains apoptotic cells in embryos of

**Table 1.** Summary of all clones analysed in both embryo and L3 of the same individual

| | Clone # 1 | | Clone # 2 | | Clone # 3 | | Clone # 4 | | Clone # 5 | | Clone # 6 | | Clone # 7 | | Clone # 8 | | Clone # 9 | | Clone # 10 | |
|---|---|---|---|---|---|---|---|---|---|---|---|---|---|---|---|---|---|---|---|---|
| | E | L | E | L | E | L | E | L | E | L | E | L | E | L | E | L | E | L | E | L |
| −I | 2 | 2 −I | 2 | 2 −I | 4 | 4 −I | 2 | 2 −I | 4 | 4 −I | 2 | 2 −I | 1 | 1 −I | 4 | 4 −I | 4 | 4 −I | 4 | 4 −I |
| I | 4 | 1 I'<br>3 I | 4 | 1 I'<br>3 I | 3 | 1 I'<br>2 I | 3 | 1 I'<br>2 I | 4 | 2 I' *<br>2 I | 4 | 1 I'<br>3 I | 3 | 1 I'<br>2 I | 4 | 1 I'<br>3 I | 4 | 1 I'<br>3 I | 4 | 2 I' *<br>2 I |
| II | 4 | 4 T1 | 3 | 3 T1 | 3 | 3 T1 | 3 | 3 T1 | 2 | 2 T1 | 2 | 2 T1 | 4 | 4 T1 | 2 | 2 T1 | 4 | 4 T1 | 2 | 2 T1 |
| III | 1 | 1 III | 2 | 2 III | 2 | 2 III | 3 | 3 III | 1 | 1 III | 4 | 4 III | 1 | 1 III | 2 | 2 III | 1 | 1 III | 2 | 2 III |
| IV | 4 | 3 IV'<br>1 IV | 4 | 2 IV'<br>2 IV | 3 | 1 IV'<br>2 IV | 2 | 2 IV' | 4 | 3 IV'<br>1 IV | 1 | 1 IV' | 4 | 2 IV'<br>2 IV | 3 | 1 IV'<br>2 IV | 3 | 1 IV'<br>2 IV | 3 | 2 IV'<br>1 IV |
| V | 2 | 2 T2 | 2 | 2 T2 | 3 | 3 T2 | 2 | 2 T2 | 2 | 2 T2 | 3 | 3 T2 | 3 | 3 T2 | 2 | 2 T2 | 2 | 2 T2 | 2 | 2 T2 |
| VI | 2 | 2 VI | 1 | 1 VI | 1 | 1 VI | 2 | 2 VI | 1 | 1 VI | 1 | 1 VI | 2 | 2 VI | 1 | 1 VI | 2 | 2 VI | 2 | 2 VI |
| VII | 5 | **5 VII** | 2 | 2 VII | 2 | 2 VII | 1 | 1 VII | 1 | 1 VII | 2 | 2 VII | 4 | 4 VII | 1 | 1 VII | 1 | 1 VII | 2 | 2 VII |
| +VII | 2 | 2 +VII | 1 | 1 +VII | 2 | 2 +VII | 2 | 2 +VII | 2 | 2 +VII | 4 | 4 +VII | 4 | 4 +VII | 4 | 4 +VII | 2 | 2 +VII | 5 | 5 +VII |

The row headers at the left show the line of cells in the embryo from which each clone originates. The numbers of cells in each clone are shown in arabic numerals. The locations of these same cells (within the lines of cells) in the larva are highlighted in bold roman numerals. Cells in lines I and IV of the embryo contribute to two separate lines of cells in the larva; within the denticulate region, apart from lines I and IV, each embryonic cell contributes to only one line of cells in the larva. Cells in line −I in the embryo never make denticles in embryo or larva, but we do not know if they contribute to two lines of cells in the larva. However, since the P compartment is made of two lines of cells in the embryo and four lines of cells in the larva, one extra row has to come from somewhere and line −I is the obvious suspect. (*) Only one these two cells showed denticles in the larva.

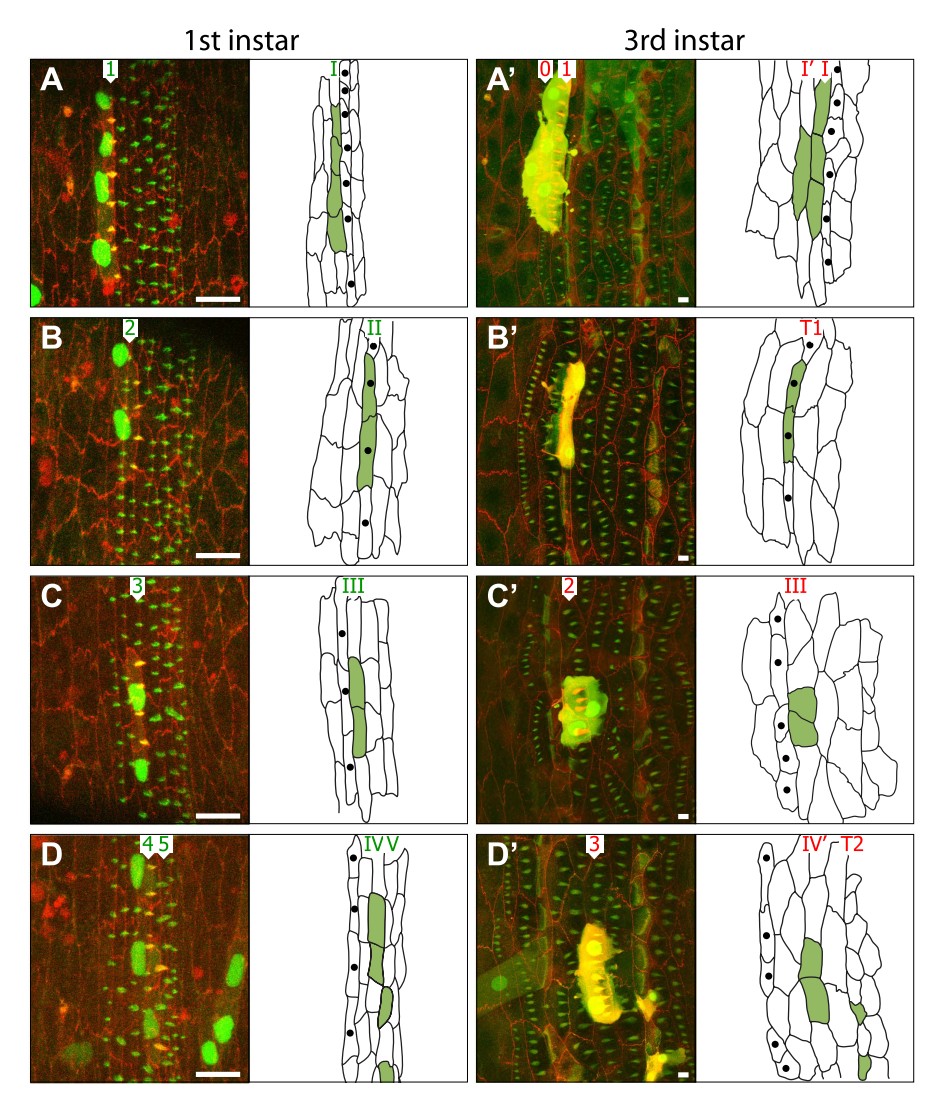

**Figure 3**. Cells rearrange and change both identity and polarity during larval development. (**A–D**) Four individuals are shown; on the left as embryos (**A–D**) while the right column shows the same four individuals as pre-L3 larvae (**A′–D′**). The drawings indicate the disposition of the cells of each marked clone (apical profiles filled in green) in the individual embryo and larva respectively. Clones were induced with *sry*.FLP in the blastoderm stage, studied in the later embryo in the pre-L1 stage and then revisited in the pre-L3. The cells of the clones are marked with cherry::moesin (red cell membranes and pre-denticles), stinger::GFP and Cd8::GFP. All the cells, clone and non-clone are marked with utrp::GFP (pre-denticles are labelled in green) and DE-cad::tomato (cell outlines labelled in red). Numbers I–VII indicate the lines of cells while numbers 1–7 mark the rows of pre-denticles in the embryo (green digits) and the larva (red digits). The cells labelled with single black dots are the T1 tendon cells. Scale bars are 10 µm. (**A–A′**) Clone of four cells. In the embryo, cells of the clone mark pre-denticles of row 1. In the larva (**A′**), cells mark pre-denticles of both row 0 (1 cell) and row 1 (3 cells). (**B–B′**) Clone of two cells. In the pre-L1 embryo, cells in the clone mark pre-denticles of row 2. In the pre-L3 larva (**B′**), the same cells are the tendon cells. No pre-denticles are marked in the larva. (**C–C′**) Clone of two cells. In the embryo, cells in the clone mark pre-denticles of row 3 while in the larva the same two cells mark pre-denticles of row 2. (**D–D′**) Four marked cells in the denticle belt. In the embryo, two cells mark pre-denticles of row 4 and in the larva these same cells mark pre-denticles of row 3. In the embryo two cells make pre-denticles of row 5 and are, presumably, tendon cells. In the larva these same cells make frank T2 tendon cells. Note the tendon cells are small with smaller nuclei, presumably of lower polytenic values than the epidermal cells. Note in Figure **D′** that there is a muscle from the adjacent more anterior segment labelled with utrp::GFP and that this muscle attaches to a T1 cell, the most anterior cell of the segment—exactly as in the adult (see *Krzemień et al., 2012*). See further cases of clones in *Figure 4*.

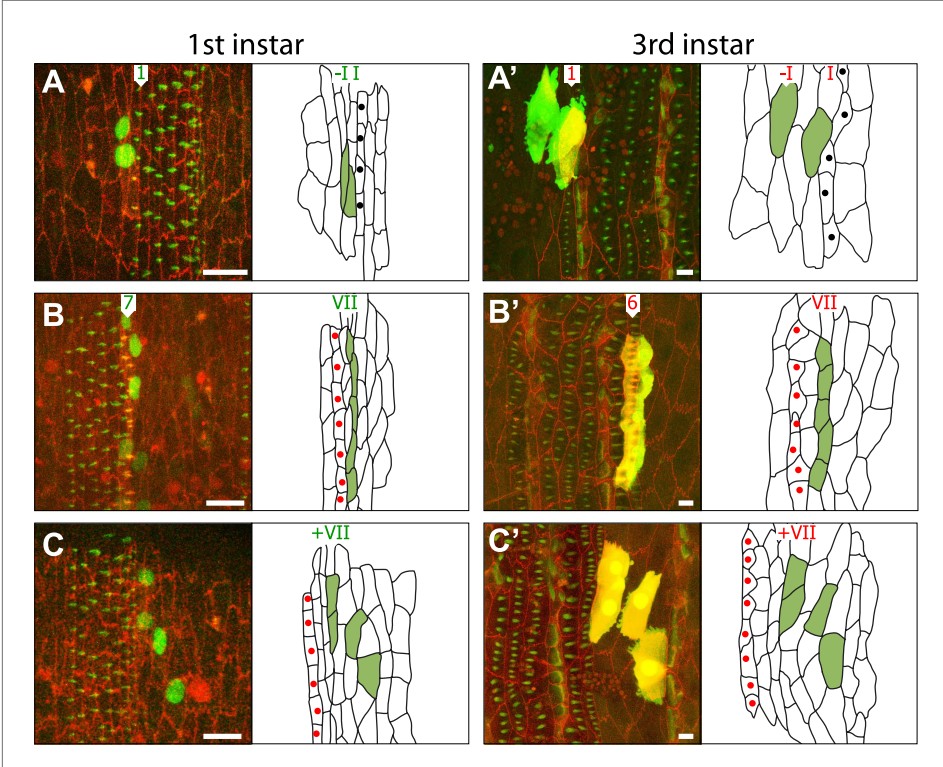

**Figure 4**. Some informative clones shown in embryo and pre-L3. (**A**) A two-celled clone in lines–I and I of the embryo. Only the larval cell I exhibits pre-denticles. There is a line of cells I' (unlabelled) in between lines–I and I in the larva. (**B**) A five-celled clone in line VII, it makes row 7 denticles in the embryo and row 6 denticles in the larva. (**C**) A four-celled clone that is in the naked cuticle posterior to line VII; this clone is not stretched out in the mediolateral axis, as the denticulate clones are. T1 tendon cells are labelled with black dots and T2 with red dots.

stage 15 as previously reported (*Bardet et al., 2008*) and we found no cell death in the epidermis of L1 or L3 larvae (data not shown).

Single clones never extended across the ventral midline that divides left from right, presumably because any clone begins with a single cell that can only be on one side of the presumptive mesoderm. Remarkably, in the embryonic denticle belts, the constituent cells of every one of those clones were arranged in a line, parallel to the mediolateral axis. Note also that the cells in the denticulate area are strongly elongated in the mediolateral axis and closely packed. However, clones that were located in the naked cuticular region between the denticle belts tended to be much less aligned. Also, the cells in the naked area, while still somewhat elongated in the mediolateral axis, were more isodiametric than those in the denticulate regions (*Figures 2E and 4C*). Observations on the embryo do not appear to show any preferred orientation in the mitoses (*Campos-Ortega and Hartenstein, 1997*; *Pfeiffer et al., 2000*). Perhaps the denticulate cells become aligned after they have been formed and, if so, it is intriguing that this movement is localised to only part of the segment.

Clone tracking demonstrated that the epithelial cells shift during postembryonic development and change their neighbours. This was particularly clear in clones within lines I and IV. Look, for example at *Figure 3A*: in the late embryo the clone consists of 4 cells belonging to line I, they are arranged side by side along the mediolateral axis and all form row 1 denticles, pointing forwards. Later, in L3, we see the same clone of four cells but now one cell has stepped out of line relative to the remaining three; this cell now becomes part of line I' and makes denticles of row 0. In every case any shift in the cells was anteriorwards: for example the cells from line IV in the embryo contributed about equally to denticle rows 3 and 4 in the L3 larva (*Figures 3D and 5*)—observations arguing that line IV cells in the embryo are the source of the extra line of cells (IV') that is found between the two rows of muscle attachments in the larva. Line III, VI and VII do not rearrange but they do move forwards relative to the pattern (see 'Cells change both identity and polarity'). The other lines of cells are stable: cell lines II and

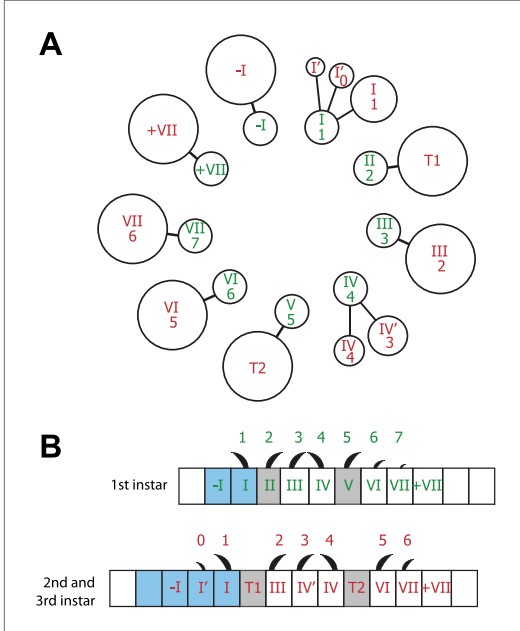

**Figure 5**. Summary of the embryo-to-larva transition. (**A**) Diagram showing the studied 9 lines of cells (green roman numerals) and rows of denticles (green arabic numerals) in the embryo and the rows of larval cells and denticles they produce in the larva (numerals in red). Both denticle rows 0 and 1 in the larva arise from line I cells of the embryo. Also, a small percentage of these line I cells contribute to line I' but make no denticles. Both rows 3 and 4 of the larva come from line IV cells in the embryo. II and V lines of cells are the tendon cells responsible for muscle attachments in the embryo that make tendon cell lines T1 and T2 in the larva. Cells of line VI and VII in the embryo produce denticles of row 5 and 6 respectively in the larva. The diameter of the circles indicates the proportion of the cells involved, for example 30% of all line I cells contribute to row 0 in the larva. (**B**) The changes between embryo and larva. Lines of cells both in front of and behind the denticles are shown. Cells shown in blue belong to the posterior compartment. Some cells of lines I and IV cells move anteriorly to form lines I' and IV', respectively. Line VII cells produce row 6 denticles in the larva.

V are the tendon cells, sites of muscle attachments in the embryo; in the larva they continue to be muscle attachments (*Figure 3B,D*). Taken together, as summarised in *Figure 5*, the clones show that, of the cells producing denticles, many change their positions relative to the pattern as they develop from embryo to larva. Also it is clear that all cell lines do not contribute equally to convergent extension, some cell lines turn into two cell lines in the larva, others make only one.

Clones within the naked cuticle behave differently; first, they are usually roundish or elongated in the anteroposterior axis (not in the mediolateral axis as are the denticulate clones); second, the clones do not change much in overall shape between the embryo and L3. However, any changes of fate would be difficult to detect as we do not have useful markers for cell fate in that region of the cuticle. The measurements mentioned above argue that there is also cell rearrangement in the naked cuticle. The only useful signpost is the row of tendon cells present near the posterior end of the A compartment.

## Cells change both identity and polarity

We have seen that individual epithelial cells can contribute to one row of denticles in the embryo and to another in the larva. Nearly 30% of line I cells, which all form row 1 denticles in the late embryo and L1, moved forward to contribute to line I' and most of these produced row 0 denticles in L2 and L3 (*Figures 3A and 5*; *Table 1*). Are all the row 0 denticles made by cells that originated from line I in the embryo? To test we looked at those line I' larval cells that do not make denticles. It is possible that any denticle-free gaps in the larval row 0 could be due to line I' cells in the larva, if they originated from line –I in the embryo. We therefore checked the fate of 29 line –I cells and asked if any went on to make denticles of row 0 (or row 1) in L2 and L3. None of these cells made denticles in the larva and none joined line I' in the larva (*Figures 5 and 4A*; *Table 1*). It appears therefore that all the row 0

denticles in line I' of the larva originate from cells that made row 1 denticles; they appear to 'remember' their embryonic propensity to form denticles, even though the larval denticles they make are distinct from those of row 1. Even so, of 12 cells originating in line I (that made row 1 denticles in the embryo and became line I' in the larva), 2 failed to make denticles in the larva. So not every cell 'remembers'.

The most striking evidence for change of identity came from line IV cells. Some individual line IV cells made row 4 denticles in the L1 stage but row 3 denticles in L2 and L3 (*Figures 3D and 5*). Row 4 denticles point forwards and row 3 denticles point backwards. These cases prove that an identified epidermal cell, even as it grows in size and increases its DNA content without division, can undergo a change in overt polarity during normal larval development. In an experimental situation involving ectopic expression of Dachsous, it was possible to change the polarity of epidermal cells as they developed through the larval stages (*Repiso et al., 2010*), but a change of polarity as a normal part of development has neither been suspected nor found before.

Regarding the other lines of cells: cells from lines III, VI and VII were found to produce distinct types of denticles at different stages of development: line III cells made only row 3 denticles in L1 but all made row 2 denticles in L3 (*Figures 3C and 5*). Similarly cells from lines VI and VII that made rows 6 and 7 denticles in the embryo made rows 5 and 6 denticles in the larva (*Figures 3C, 4 and 5*). Thus their identities change and with a particular bias; for they all make more anterior pattern elements in the larva than they did in the embryo. Cells from lines II and V make row 2 and 5 denticles in the embryo but they do not make denticles at all in the larval stages (*Figures 3B,D and 5*). In so far as denticle position, size and polarity are tokens of cell identity, then these observations prove that single cells change their identity during normal development. Furthermore, the results argue for the existence of supracellular systems that act continuously to maintain a stable cuticular pattern in spite of considerable rearrangement of the cells (*DiNardo et al., 1994*; *Heemskerk and Dinardo, 1994*; *Alexandre et al., 1999*).

## How do tendon cells stop producing denticles?

In order to maintain the same denticle pattern, the cells must compensate for changes in the behaviour of the tendon cells: we have shown that, although the embryonic tendon cells produce denticles (*Hatini and DiNardo, 2001*), the larval tendon cells do not. The instruments of this change are unknown. It may be relevant however that the expression of the *stripe* (*sr*) gene, that drives the differentiation of tendon cells, has two isoforms expressed at different times (*Volohonsky et al., 2007*). The embryonic tendon cells express the B isoform of *sr*. Later, in the pre-L1 stage, when the muscles have already attached to the tendon cells and the pre-denticles have been formed, the tendon cells begin to express also the *srA* isoform (*Frommer et al., 1996*; *Becker et al., 1997*; *Vorbruggen and Jackle, 1997*). Perhaps the expression of *srA* could change the nature of the tendon cells? To test we expressed *srA* and *srB* prematurely and ectopically in the early embryo in clones and found that the early expression of either of these forms failed to block the formation of pre-denticles and denticles in L1. However, in the larva, cells in those clones over-expressing either SrA or SrB acquired the actin palisades characteristic of muscle-attached tendon cells in L3 (*Figure 1B,D*) and did not form any denticles (*Figure 6*). One simple interpretation is that, although *sr* expression allows denticle formation in the embryo, it blocks denticle formation in the larva.

## How plastic are the cell identities?

Our results do not argue for complete lability in cell identities. For example, it would be unexpected, if, in normal development, a cell were to change from A to P identity. Indeed there is no case of cells producing row 2 (A compartment) in the embryo that make row 1 denticles (P compartment) in the larva, even though this would only require a small anteriorwards shift in cell fate, as happens to other lines of cells.

## How to define PCP when polarities can change?

Our results raise queries about the stability of PCP. What do we conclude if the polarity of a cell changes during development? In *Drosophila* many experiments suggest that an individual cell's polarity is not imposed by pervasive forces but depends, locally, on its interactions with its neighbours. For example, wild type cells that are adjacent to a nascent clone of cells lacking the *frizzled* gene form hairs of reversed polarity (*Gubb and Garcia-Bellido, 1982*). Now we report changes, during normal development, in the polarity of larval cells that do not divide. In both cases it is likely that polarity of a cell is driven by interactions with contacting cells. During development from embryo to larva, we have shown that cells, as a consequence of rearrangement, acquire new neighbours. Moreover, if these new neighbours were to present different amounts of Dachsous or Fat on their abutting membranes, this would lead to different arrangements of heterodimers and consequent polarity changes (*Ma et al., 2003*; *Casal et al., 2006*; *Repiso et al., 2010*).

## Row nomenclature

Our kind reviewers have asked us to review how we name the denticle rows and to consider alternative nomenclatures. The naming of the rows is done objectively in both L1 and L2/3 but a comparison between the stages is open to various interpretations. Comparison is problematic because we have shown that two lines of cells that make denticles in L1 do not make denticles in L2/3. Since the number of denticle rows is about the same in all three larval stages, it follows that the two replacement denticular rows in L2/3 must be made by cells that either did not have denticular identity in L1, or by cells that contributed to different denticle rows in L1.

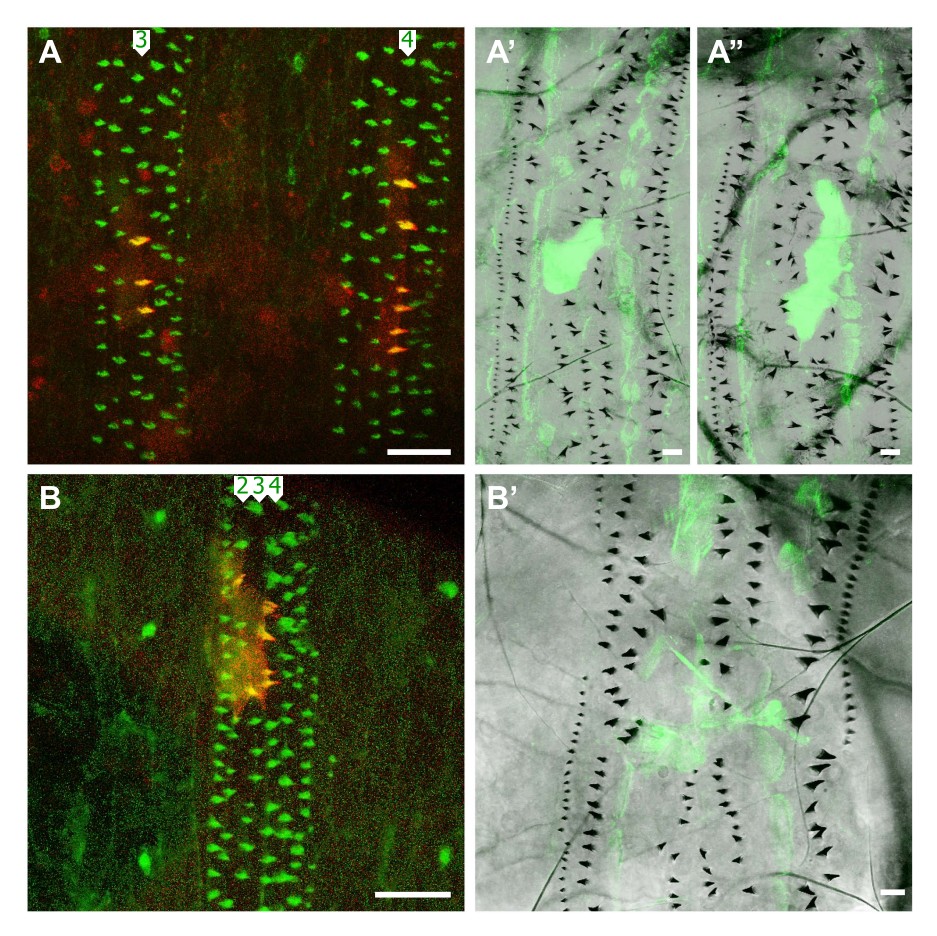

**Figure 6**. Clones expressing the *stripe* gene block L2 and L3 denticle formation. (**A**–**A"**) Two clones (labelled with cherry::moesin and Cd8::GFP) in adjacent segments in one individual make denticles in pre-L1 but no denticles in L3. The B isoform of *sr* is expressed in these clones which are found in the row 3 of the fourth and the row 4 of the fifth abdominal segments. (**B**–**B'**) One clone expressing the A isoform of *sr* in the embryo makes denticles in rows 2 to 4 of the sixth abdominal segment, but makes no denticles in the larva. The denticle rows are disturbed in this case.

First, we name the denticle rows according to their position in the series of rows, with some input from interpretation. For example, in L2/3, we name the extra row at the front, row 0 because of its position *vis-à-vis* the compartment border and because it is one row anterior to row 1—row 1 is itself the most posterior row in the P compartment in all larval stages and was named long ago and extensively studied in the developing L1 (**Dougan and DiNardo, 1992**). Row 2 is so named because it is the next row after row 1. Row 2 in L1, as we and others (**Dilks and DiNardo, 2010**) have described, is made by the tendon cells, T1. But we show here that, in L2/3, the second row of denticles we now call row 2 is made by the cells lying posterior to T1, and the clones show indubitably that the cells that make this row in L2/3 are different from those that made row 2 in L1. Row 3 is named as the next row back from row 2 in both L1 and L2/3 and and the other rows in sequence, up to and including row 7. Second, the rows are named because of their characteristics. In L1, the rows differ in polarity and also in structure. In L1, most of the denticles are similar, although variable, sizes, but row 5 has clearly larger denticles. However in L2/L3 the rows are more individually sized and can be distinguished on those grounds (e.g., row 3 denticles are small and row 5 denticles particularly large, see *Figure 1*). In addition we name the rows because of their polarity; the denticles of rows 0, 1 and 4 point forward in all larval stages.

The problems arise when one compares row numbers in L1 with those in L2/3: for example one reviewer questioned our nomenclature and proposed that maybe row 2 of L1 is not made in L2/3, so what we are calling row 2 in L2/3 is more homologous to row 3 (of L1) than row 2 (of L1). But if we were

to rename our row 2 'row 3' in L2/3 (as he/she suggests) then we would have to rename the more posterior rows and if we were to do that then 'row 5' in L2/3 would point forwards (as row 4 does in all larval stages in our nomenclature) and 'row 6' in L2/3 would have the largest denticles (like row 5 in all larval stages in our nomenclature). These inconsistencies argue against using the reviewer's alternative nomenclature; in our view all the characteristics of the different denticle rows should be used as objectively as possible to help name them.

### Evolution and the conservation of pattern

The functional outcome of the developmental processes we describe is the segmented pattern of denticle bands. This pattern appears to be important for crawling (*Dixit et al., 2008*) and therefore, presumably, for function. If so, it needs to be conserved not only during the evolution of species (*McGregor et al., 2007*) but also in all the three larval stages of each species. However, conflicting with this need, is the fundamental and atavistic process of convergent extension that, necessarily, causes cell rearrangement. This conflict is resolved by the use of supracellular mechanisms, mechanisms that can build a fixed pattern in a shifting and labile cell population (*DiNardo et al., 1994*; *Heemskerk and Dinardo, 1994*; *Alexandre et al., 1999*). To achieve the same final pattern in spite of cell rearrangement it becomes necessary to drive, as we have found, changes in individual cell identity. We believe that the *Drosophila* embryo is exceptional because, throughout the growth of these *Diptera*, the epidermal cells are neither dividing nor dying. This makes our finding that individual cells become reallocated to different fates and polarities particularly eloquent.

The ability of developmental processes to conserve pattern in a changing cellular constitution is emphasised when related species are compared. In classic work with the development of the vulva in nematode species, (*Eizinger and Sommer, 1997*; reviewed in *Sommer (1997)*) showed how two similar species build a common structure by contrasting molecular and cellular mechanisms. Illustrating again that the pressure of selection acts directly on the outcome but only indirectly on the mechanisms of development.

## Materials and methods

### Mutations and transgenes

Flies were reared at 25°C in standard food. The Flybase (*Marygold et al., 2013*) entries of the relevant constructs used in this work are the following: *en.GAL4*: *Scer/GAL4^en-e16E^*; *sr.GAL4*: *sr^md710^*; *DE-cad::GFP*: *shg^Ubi-p63E.T:Avic\GFP-rs^*; *DE-cad::tomato*: *shg^KI.T:Disc\RFP-tdTomato^*; *sqh.utrp::GFP*: *Hsap\UTRN^sqh.T:Avic\GFP-EGFP^*; *UAS.cherry::moesin*: *Moe^Scer\UAS.P\T.T:Disc\RFP-mCherry^*; *UAS.stinger::GFP*: *Avic\GFP^Stinger.Scer\UAS.T:nls-tra^*; *UAS.srA*: *sr^A.Scer\UAS^*; *UAS.srB*: *sr^b.Scer\UAS^*. *tub>stop>GAL4*: *P{GAL4-αTub84B(FRT.CD2).P}*; *UAS.Cd8::GFP*: *Mmus\Cd8a^Scer\UAS.T:Avic\GFP^*. *UAS.Apoliner*: *P{UAS-Apoliner}5*. *Act.GAL4*: *P{Act5C-GAL4}17bFO1*.

### Experimental genotypes

> (*Figure 1*) *w; DE-cad::tomato/en.GAL4; UAS.cherry::moesin/utrp::GFP*.
> (*Figure 1*) *w; DE-cad::tomato, utrp::GFP; sr.GAL4/UAS.cherry::moesin*.
> (*Figure 2*) *w; DE-cad::tomato utrp::GFP/CyO*.
> (*Figure 2*) *w; DE-cad::GFP/utrp::GFP*.
> (*Figure 3*, *Figure 4*) *w; tub>stop>GAL4 UAS.GFP utrp::GFP/DE-cadherin::tomato UAS.stinger::GFP;UAS.cherry::moesin/sry.FLP*.
> (*Figure 6*) *w; tub>stop>GAL4 UAS.cd8::GFP utrp::GFP/UAS.SrA (or UAS.SrB); UAS.cherry::moesin/sry.flp*.

### Clone induction

A transgene expressing FLP specifically at the blastoderm stage was created by placing the *sry-alpha* promoter (−331 to +130) (*Schweisguth et al., 1989*) upstream of a cDNA encoding FLP (*Golic and Lindquist, 1989*). Transgenic flies were obtained by P-element transformation. Several such transgenes were recombined with *tub>stop>GAL4* and *P{UAS-GFP^S65T^.nls}* (*Neufeld et al., 1998*) and a combination that gave a useful frequency of clones was selected for further work. The *sry-alpha* promoter is only expressed for about 15 min during cellularisation (*Schweisguth et al., 1990*). Clones are therefore expected to be induced before or after the first blastoderm division, which follows cellularisation. Since most clones comprised 4 cells and since epidermal cells divide three times on average

(*Vincent and O'Farrell, 1992*; *Foe and Alberts, 1983*), it appears that clones were mostly induced soon after the first blastoderm division.

Stage 15 embryos with clones of 1–4 cells in the abdominal segments A2–A6 were mounted in a drop of Voltalef 10S oil on a microscope slide and imaged using a Leica SP5 confocal microscope. The embryos were subsequently removed, kept at 25°C on an agar plate with fresh yeast paste and reexamined 48–60 hr later (i.e., in pre-L3 or early L3 stages). To identify the lines of cells labelled, we used a combination of features: the number of rows of pre-denticles, the localisation of the tendon cells and the distinct types of pre-denticles specific for each row. For example, some III cells that did not completely align with each other were still scored as III cells based on their position relative to the tendon cells and also on the size and number of their pre-denticles—usually in the larva, III cells make about four big pre-denticles, whereas cells of line IV tend to form more denticles per cell, but of smaller size.

## Quantification of cells in embryo and larva

A quadrilateral *abcd* was drawn using as reference the ventral papillae *a'b'c'd'* whose sides *ac* and *bd* line up with the P/A boundary between the third and fourth and the fourth and fifth abdominal segments respectively (*Figure 4B*). Around 10 straight lines were then drawn between and perpendicular to the lines *ac* and *bd*. The number of cells intersected by these lines was counted and the average of the ten lines used to estimate the number of cells in the A/P axis (*Figure 4F*). We also measured separately the number of cells in the denticular region (rows 1 to 7 in the embryo and 0 to 6 in the larva) as well as the cells forming the naked cuticle between two denticle belts (comprising the posterior half of the A compartment plus the anterior half of the P compartment in the same segment). For the total number of cells we counted the cells that were, entirely or partially, inside the quadrilateral. For statistical analysis we used the R programming language and software environment (*R Core Team, 2013*).

## Acknowledgements

We thank Marco Antunes, Marcus Bischoff, Paola Cognini, Joanna Krzemień, Thomas Lecuit, Eurico Morais de Sá, and the Bloomington Stock Center for flies, and Charles Girdham for his help in making *sry.FLP* construct. We also thank Gary Struhl for advice and the anonymous reviewers for comments. PS had a fellowship from the Fundação para a Ciência e a Tecnologia and a studentship from the Cambridge Philosophical Society.

## Additional information

### Funding

| Funder | Grant reference number | Author |
| --- | --- | --- |
| Wellcome Trust | WT086986MA | Peter A Lawrence |
| Wellcome Trust | WT096645MA | Peter A Lawrence |

The funder had no role in study design, data collection and interpretation, or the decision to submit the work for publication.

### Author contributions

PS, Conception and design, Acquisition of data, Analysis and interpretation of data, Drafting or revising the article; J-PV, Conception and design, Drafting or revising the article, Contributed unpublished essential data or reagents; IMP, Conception and design, Analysis and interpretation of data; PAL, JC, Conception and design, Analysis and interpretation of data, Drafting or revising the article

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
