## [Decision Letter]

Thank you for sending your work entitled “Plasticity of both planar cell polarity and cell identity during the development of *Drosophila*” for consideration at *eLife*. Your article has been favorably evaluated by a Senior editor and 3 reviewers, one of whom is a member of our Board of Reviewing Editors.

The Reviewing editor and the other reviewers discussed their comments before we reached this decision, and the Reviewing editor has assembled the following comments to help you prepare a revised submission.

Overall, the reviewers felt that the elegant technique and the demonstration that cells change their morphology during different stages of development was of broad interest. The technique is clever – they induced clones with sryFLP and marked cells of clones with *cherry::moesin*, *singer::GFP* and *Cd8::GFP*. All cells were marked with *utrp::GFP* and *DE-cad::tomato*. This allows an excellent resolution of cell outlines and following clone fates. Using this approach for clone tracking, they discovered that epithelial cells shift during postembryonic development, and that cells shift anterior wards. Their finding that a row of cells repolarizes is quite interesting.

However, there were several points that require clarification, and more experiments needed to support (or modify) some of the authors’ conclusions.

1) The authors state that the number of cells in a defined portion of a segment remains constant at all three stages, which they say confirms that epidermal cells do not divide or die. While this might be the case, this observation does not prove that mechanism – it could also be true that some cells die and are replaced. The manuscript would be improved if some evidence such as caspase staining or BrdU incorporation was added. More information about clones between the denticle belts could also be informative. The authors may already have this data. The assumption that there is no division is based on 83 clones that are largely within the denticle belts – do clones between the belts behave the same way? The evidence for CE is based upon the increased number of cells rows (presumably between the denticle belts) from embryo to larva. This could occur through cell division, but if a significant number of clones between the denticle belts show no change in cell number, then the evidence for CE would be stronger

Also, clones between the denticle belts would be expected to change shape between embryo and larva if CE occurs – could this analysis be done? The major shape change in the ventral epidermis seems to take place in the late embryo (Figure 2, but it is the change from late embryo to larva (Figure 2 i.e. quantified: why?

2) While row IV cells clearly change their polarity (when they form row IV'; see schematic Figure 5), it is less clear if other rows clearly change their polarity or fate. For example, embryonic row 2 will form the larval T1 cells and row III cells will form denticle belt 2, which the authors consider a fate change. An alternative interpretation is that embryonic denticle belt 2 is lost (i.e., tendon cells simply stop to make denticles after the molt) and what appears as larval denticle belt 2 is actually embryonic (and larval) denticle belt 3. Similarly, larva could just stop having a denticle belt 5 (T2 cells) and what looks like larval belts 5 and 6 could actually be larval belts 6 and 7 (i.e., belt 5 ceases to exist). This alternative model should be discussed.

3) The authors observe cells changing fate from row 4 to row 3 between L1 and L2 instars resulting in a reversal of denticle orientation. They state that a ‘change of polarity as a normal part of development has neither been suspected nor found before’. However, earlier in the manuscript they describe a (previously reported) change in polarity during normal embryonic development in row 1 and 4 cells, as predenticles point to the posterior, but mature denticles point to the anterior. There seems to be a contradiction here that the authors should address.

4) The studies on SrA and SrB need more links to either upstream regulation or downstream targets in order to be informative as to the mechanisms underlying the changes in cell identity. The ‘supracellular systems’ underlying the proposed global re-patterning between larval stages are not investigated, nor (perhaps surprisingly) is any model proposed. This could be considered a shortcoming of the manuscript, and should at least be addressed in the text.

5) A problem in this field at present is the inconsistent definition of PCP in denticle cells. It has been defined by both predenticle location and denticle hook orientation. However, it seems that these two properties are not tightly coupled either during normal development or in experimental situations, and so cannot substitute for each other. This situation contrasts with the *Drosophila* wing, for example, where prehair location and hair orientation seem tightly coupled. Changes in hair polarity in wing cells appear to be consistently preceded by changes in prehair location. This difference makes comparisons between findings on PCP in the embryo/larva with those in adult tissues problematic. I recommend that the authors clarify which specific aspect of cell polarity (denticle orientation, predenticle location) they are discussing, rather than just referring to ‘cell polarity’ or ‘PCP’.

6) One view of PCP refers to a cell’s perception of its orientation within a tissue, that is, where it ‘thinks’ anterior, posterior etc are located. Therefore, when a cell’s PCP is altered, it is either because it no longer knows where anterior/posterior are, or, for example, it has confused anterior with posterior. In this worldview, the cell fate change from a row 4 cell to a row 3 cell does not appear to be a change in PCP (despite the change in denticle orientation), as the cell seems always to know the correct direction of anterior and posterior. Please address this in the revised manuscript.

7) An unrelated point that the authors might discuss is where all the cells in larval rows 3 and 4 originate. It appears that the population of cells within one embryonic row normally makes a single larval row. The exception is embryonic row 4, which generates both larval rows 3 and 4. Without cell division, how is a single embryonic row able to populate two larval rows? Cell numbers between larval rows do not appear (by eye) to differ greatly. Is there another source of cells for larval rows 3 and 4?

[Editors' note: further clarifications were requested prior to acceptance, as described below.]

Thank you for resubmitting your work entitled “Plasticity of both planar cell polarity and cell identity during the development of *Drosophila*” for further consideration at *eLife*. Your revised article is now acceptable except for a minor addition to the Discussion. All of the reviewers felt that the Discussion should at least add a paragraph addressing/dismissing the alternative model raised by the reviewers (even if that interpretation is not favored by the authors in the text).

*Reviewer 1 comments*:

My main concern with the authors’ revisions is that they do not adequately address the alternative interpretation of their data proposed by the reviewers in point (2). The authors state that the reviewers’ model is ‘inviable’; however, I believe that their own interpretation has some problems, and the reviewers’ model may be the more parsimonious. From the paper: “line III cells made only row 3 denticles in L1 but all made row 2 denticles in L3. Similarly cells from lines VI and VII that made rows 6 and 7 denticles in the embryo made rows 5 and 6 denticles in the larva. Thus their identities change and with a particular bias; for they all make more anterior pattern elements in the larva than they did in the embryo”. So the authors show that line III cells make row 3 denticles in L1, but row 2 denticles in L3. They call this a change in identity, presumably because ‘row 3’ cells have become ‘row 2’ cells (the name has changed). However, in both L1 and L3, line III cells are two cells posterior to the compartment boundary, immediately posterior to a tendon cell and produce a posterior pointing denticle. How is the identity changed? In what way have the cells taken a ‘more anterior’ identity? Similar questions can be asked about the identity of Line VI and VII cells.

I think the distinction between the authors’ view and the reviewers’ model is important. The authors’ view, in which most cells within the denticle belt change to a more anterior identity between L1 and L3, would indeed suggest the involvement of supracellular mechanisms that can redefine cell identity across the denticle belt. In contrast, the reviewers’ model would only require local changes in cell morphology, for example, tendon cells stop producing denticles, that would probably not require supracellular patterning mechanisms. I think the authors should discuss the reviewers' model further rather than labeling it inviable.

*Reviewer 2 comments*:

I am still not convinced that some of the cells really change their fate rather than just stop making denticles. It looks to me as if larval denticle belts 5 and 6 thus could actually be larval belts 6 and 7 (i.e., belt 5 ceases to exist). Although the authors consider this idea ‘inviable’, it is, in my view, plausible (I think I understand the power of cell tracking using clones). Perhaps this discussion is semantic and due to different ideas of cell fate (i.e., is any change of how a cell behaves at two different times of development a change in cell fate?). Of course, I agree with the authors that the case for row IV' cells that revert their polarity with respect to the A/P axis is entirely different.

---

## [Author Response]

*1) The authors state that the number of cells in a defined portion of a segment remains constant at all three stages, which they say confirms that epidermal cells do not divide or die. While this might be the case, this observation does not prove that mechanism – it could also be true that some cells die and are replaced*.

Regarding this point, which we found very constructive, we should reiterate that the *Calypterate* larval epidermis is unusual. If there were divisions in it then that would be a big surprise. These cells reach a high degree of polyteny and noone has ever reported mitosis in, and they have been studied by many people who were interested, who observed, did Feulgen staining and live imaging. For example Wagner (1951) and [34] for *Calliphora*. However the reviewer does have a point in that our studies show that the cell rearrangement occurs during the first larval instar and that period has been little studied, even though it is known (34) that the cells are diploid at hatching from the egg.

However, regardless of what has gone before we submit that our own observations clinch the matter: in this paper we have followed many individual epidermal cells in the reported (and unreported) clones all the way from embryo to L3 and from around 400 cells there are only few changes in the cell number in a clone: in a couple of cases we “lost” a cell and only in five clones we “gained” cells. In both cases we suspect recording errors. This evidence is much more definitive than any previous observations.

*The manuscript would be improved if some evidence such as caspase staining or BrdU incorporation was added*.

This has been well done by others: BrdU studies are reported in Britton and Edgar (1998), they reveal that at any time most epidermal the cells are synthesising DNA; this result tells that the cells are becoming more and more polytene. No mitoses were reported.

In response to this request we have added a brief study with a new live marker for caspase We have found that the Apoliner marker stains apoptotic cells in embryos as previously reported (2) and throughout the late embryo; we observed no cell death in the epidermis of L1 or in later larval stages. In the embryos apoptosis was detected in the CNS, as expected.

It is relevant that, so as far as we know, no one has reported cell death in the larval epidermis.

We submit our method of checking for cell loss is superior to previous ones because we have followed individual identified cells throughout the whole larval period, and found all those cells survived and did not divide. As they are a fair sample of all the epidermal cells, we submit that they are telling us the answer unequivocally but see below for further work with more clones.

*More information about clones between the denticle belts could also be informative. The authors may already have this data. The assumption that there is no division is based on 83 clones that are largely within the denticle belts – do clones between the belts behave the same way*?

This suggestion is made by another reviewer below and we agree with it. We have made new experiments and accumulated such clones. We agree with the reviewers that this data strengthens the paper. We have found the clones in the naked cuticle behave somewhat differently from the ones we studied before, that is, like them they do not change in cell number but the clones we studied changed very little in shape. It would appear that the amount of rearrangement is less in this area. One problem with analysing these clones was that we have few cuticular landmarks (like the denticle rows) to use. In response to the reviewers’ points, we have measured the changes in cell dimensions in both the denticulate and the naked cuticle and the result argues that the amount of convergent extension is similar in the naked than in the denticulated region.

*The evidence for CE is based upon the increased number of cells rows (presumably between the denticle belts) from embryo to larva. This could occur through cell division, but if a significant number of clones between the denticle belts show no change in cell number, then the evidence for CE would be*
*stronger*

*Also, clones between the denticle belts would be expected to change shape between embryo and larva if CE occurs – could this analysis be done*?

See above: we have now completed these clones and they support our story.

*The major shape change in the ventral epidermis seems to take place in the late embryo (*Figure 2*), but it is the change from late embryo to larva (*Figure 2*) that is quantified: why*?

The change takes place subsequent to the secretion of the L1 cuticle (before hatching from the egg) but before secretion of the L2 cuticle; this is our finding and it is over this entire period that we have quantified the total cell numbers in a fixed domain.

*2) While row IV cells clearly change their polarity (when they form row IV'; see schematic*
Figure 5*), it is less clear if other rows clearly change their polarity or fate. For example, embryonic row 2 will form the larval T1 cells and row III cells will form denticle belt 2, which the authors consider a fate change. An alternative interpretation is that embryonic denticle belt 2 is lost (i.e., tendon cells simply stop to make denticles after the molt) and what appears as larval denticle belt 2 is actually embryonic (and larval) denticle belt 3. Similarly, larva could just stop having a denticle belt 5 (T2 cells) and what looks like larval belts 5 and 6 could actually be larval belts 6 and 7 (i.e., belt 5 ceases to exist). This alternative model should be discussed*.

The clones document what actually happens, cell by cell. So this above “alternative model” is inviable. Indeed we are not providing a “model”; we are providing a description of exactly how the cells in the clone generate the two patterns, the embryonic and the larval. There is no room for ambiguity in this; that is why the clones are so powerful. We see what each cell does in the early and late stages, as we follow each cell as it contributes first to the L1 pattern and then to the L2 or L3 pattern. In the case of the cells making the L1 denticle row 2, these cells also make muscle attachments (tendon cells) in the embryo and later in L2 and L3 these same cells remain as tendon cells but make no denticles.

*3) The authors observe cells changing fate from row 4 to row 3 between L1 and L2 instars resulting in a reversal of denticle orientation. They state that a ‘change of polarity as a normal part of development has neither been suspected nor found before’. However, earlier in the manuscript they describe a (previously reported) change in polarity during normal embryonic development in row 1 and 4 cells, as predenticles point to the posterior, but mature denticles point to the anterior. There seems to be a contradiction here that the authors should address*.

This is a good point. In the L1 the predenticles are first oriented (in rows 1 and 4) opposite to the orientation of the denticles the cells later make. This is a fact. So how to digest this fact? We have chosen to describe the polarity of the denticle as indicating PCP of the epidermal cells. We could have chosen the polarity of the predenticle but that would have been a more demanding and idiosyncratic choice, cuticular denticles and hairs have been studied in countless papers and they are the most tangible indicators of polarity. In this paper we have discussed mainly the clones; however, the cellular mechanism of predenticle and denticle formation, and how it differs in L1 from L2 and L3 is interesting, as the reviewer cleverly intuits, but is part of a longer and more detailed paper that we are now preparing for a specialised journal.

*4) The studies on SrA and SrB need more links to either upstream regulation or downstream targets in order to be informative as to the mechanisms underlying the changes in cell identity. The ‘supracellular systems’ underlying the proposed global re-patterning between larval stages are not investigated, nor (perhaps surprisingly) is any model proposed. This could be considered a shortcoming of the manuscript, and should at least be addressed in the text*.

The underlying ‘supracellular mechanisms’ that establish the pattern have been extensively studied and published by many others ever since Wieschaus and Nusslein-Volhard’s famous genetic screen. These mechanisms establish the segments in the body plan of flies and polarise and pattern the individual segments. For example Hedgehog is made by the posterior compartment of each segment and spreads in to the anterior compartments on either side where it is received by Ptc and Smo. Wg is essential for the patterning, without Wg the denticle belts do not form. In short, there are hundreds of papers on these supracellular mechanisms but they are nearly all concerned with the embryo. While there are good reasons to think that patterning is done in the L2 and L3 by the same mechanisms as the embryo, this is not established. As for the specification of the individual denticle rows there are fewer papers, but those by DiNardo and colleagues and by Alexandre et al., to which we refer, are concerned with the embryo, the development of the L1 cuticle and the mechanisms that contribute to making its 6 rows of denticles.

In addition we have published previously two papers (7; 38) on those mechanisms responsible for denticle identity and particularly polarity in the L2 and L3 stages. But these interesting matters cannot be discussed further here; they could take this relatively short paper and transform it into a long review.

*5) A problem in this field at present is the inconsistent definition of PCP in denticle cells. It has been defined by both predenticle location and denticle hook orientation. However, it seems that these two properties are not tightly coupled either during normal development or in experimental situations, and so cannot substitute for each other*.

We see this inconsistency as a fact that challenges our rather limited perspectives. This is the interest in the abdomen of the adult and the larvae; these atavistic body parts (i.e., the trunk) might make us accept we have a long way to go before we understand planar cell polarity.

*This situation contrasts with the Drosophila wing, for example, where prehair location and hair orientation seem tightly coupled. Changes in hair polarity in wing cells appear to be consistently preceded by changes in prehair location*.

What the reviewer writes here is debatable; in the wing it has been described (Strutt and colleagues) that in various clones and mutant genotypes (such as fz^−^), the prehairs are formed in the centre of the cells and subsequently the hairs are oriented specifically and in whorls. How the hairs process from the cell centre to one side has not been described. Indeed the wing has been presented as a simple system by wing advocates but in truth is also complex. Look for example at the ridges recently studied by Collier’s group. They also give a different and if you like “problematic” picture in which the ridges have a different polarity than the hairs.

*This difference makes comparisons between findings on PCP in the embryo/larva with those in adult tissues problematic*.

Quite so, Nature can be irritating in questioning our neat ways of describing her!

*I recommend that the authors clarify which specific aspect of cell polarity (denticle orientation, predenticle location) they are discussing, rather than just referring to ‘cell polarity’ or ‘PCP’*.

We have tried to clarify this, but it is tricky. I hope the paper makes clear what we are looking at (it’s always the denticles in this paper). And, in general, one could ask which indicator should be preferred? Localisation of proteins e.g., or is it better to rely on ommatidial chirality, or what about multicellular mammalian hair follicles, or perhaps early actin predenticles or maybe later cuticular protrusions? They are all indicators of polarity as they are anisotropic, in the rare cases when they are in conflict perhaps they should be teaching us something rather than perturbing us!

*6) One view of PCP refers to a cell’s perception of its orientation within a tissue, that is, where it ‘thinks’ anterior, posterior etc are located. Therefore, when a cell’s PCP is altered, it is either because it no longer knows where anterior/posterior are, or, for example, it has confused anterior with posterior. In this worldview, the cell fate change from a row 4 cell to a row 3 cell does not appear to be a change in PCP (despite the change in denticle orientation), as the cell seems always to know the correct direction of anterior and posterior. Please address this in the revised manuscript*.

We have discussed this almost vitalistic point between us, and we believe it is interesting. In the case of row 1 and row 4 we have discovered what determines the polarity difference from the other rows, but that is for the forthcoming paper we mention above.

However, there is an important general point raised by this comment. That the thousands of mosaic experiments that have been done, mostly in *Drosophila*, show that the view referred to by the reviewer, that individual cells “know” where the head is, is not correct. Polarity is determined locally by each cell interacting with its neighbours. We could not put this into the Introduction as really this paper is about the relationship between pattern and development and the changing identity (and polarity) of cells. Our paper does not address the view of PCP offered by the reviewer or ask how PCP can be defined. But the reviewer raises some interesting aspects of PCP and we have added a little general discussion of the stability of PCP.

*7) An unrelated point that the authors might discuss is where all the cells in larval rows 3 and 4 originate. It appears that the population of cells within one embryonic row normally makes a single larval row. The exception is embryonic row 4, which generates both larval rows 3 and 4. Without cell division, how is a single embryonic row able to populate two larval rows? Cell numbers between larval rows do not appear (by eye) to differ greatly. Is there another source of cells for larval rows 3 and 4*?

This comment perplexes us as the clones have told the exact answers to these questions. That is what we have been studying in the paper. The cells do not change in total number; they rearrange. As a result there are more rows in the A/P axis and fewer cells in the mediolateral axis. Another complication is that cells that make row 2 in L1 make no contribution to denticle rows in the L2; this means that the cells that make row 2 in L2 and L3 have to be found from the others, and therefore mean that the total number of rows of cells increases in the denticular region by more than the one referred to (embryonic row 4) by the reviewer. In fact the measurement shows an increase of several rows in the A/P axis due to CE (Figure 2). So it is incorrect to think that row 3 and 4 are exceptional in this respect.

[Editors' note: further clarifications were requested prior to acceptance, as described below.]

*Thank you for resubmitting your work entitled “Plasticity of both planar cell polarity and cell identity during the development of Drosophila” for further consideration at* eLife*. Your revised article is now acceptable except for a minor addition to the Discussion. All of the reviewers felt that the Discussion should at least add a paragraph addressing/dismissing the alternative model raised by the reviewers (even if that interpretation is not favored by the authors in the text) […]*.

We have read the reviewers’ opinions with interest and in response, and as requested, we have added a new section to the Discussion. This discusses the issues raised by the reviewers, specifically the nomenclature of the denticle rows and some aspects of cell identity. As we are all aware the concepts of cell identity and determination are important but their definitions are not precise; nor do these words mean the same things to different people. And the reviewers are right there are different ways one could define the row numbers and different ways one could describe the results of the clones. We hope that our attempt to discuss these issues will raise the reviewers’ points clearly enough, and present our own interpretation also, so that readers will see the several sides of this conundrum. We would be happy to credit the reviewers in this paragraph, if they wish to reveal their identity to us. As of now we must perforce refer to them without names.

We hope the reviewers will be satisfied with what we have written and we thank them for their efforts to make the paper clearer. But we also have a message for the editors: do you not think this discussion in the section we have added (“row nomenclature”) is somewhat arcane, and would not that section be better placed in the discussion between authors and reviewers that you also publish?

We have not changed the diagram in Figure 5 because the purpose of that diagram is to show the relationships between the lines of cells that make the denticle rows in L1 and the lines of cells that make the rows in L2/3. As we don’t know how the naked cells are related to each other in the different larval stages, we cannot fill in the diagram for the naked cells as we have been asked to do.